# GUIDING ENERGY-BASED MODELS
# VIA CONTRASTIVE LATENT VARIABLES

**Hankook Lee**[A*]  **Jongheon Jeong**[B]  **Sejun Park**[C]  **Jinwoo Shin**[B]
[A]LG AI Research  [B]KAIST  [C]Korea University
hankook.lee@lgresearch.ai, {jongheonj, jinwoos}@kaist.ac.kr, sejun.park000@gmail.com

## ABSTRACT

An energy-based model (EBM) is a popular generative framework that offers both explicit density and architectural flexibility, but training them is difficult since it is often unstable and time-consuming. In recent years, various training techniques have been developed, *e.g.*, better divergence measures or stabilization in MCMC sampling, but there often exists a large gap between EBMs and other generative frameworks like GANs in terms of generation quality. In this paper, we propose a novel and effective framework for improving EBMs via contrastive representation learning (CRL). To be specific, we consider representations learned by contrastive methods as the true underlying latent variable. This *contrastive latent variable* could guide EBMs to understand the data structure better, so it can improve and accelerate EBM training significantly. To enable the joint training of EBM and CRL, we also design a new class of latent-variable EBMs for learning the joint density of data and the contrastive latent variable. Our experimental results demonstrate that our scheme achieves lower FID scores, compared to prior-art EBM methods (*e.g.*, additionally using variational autoencoders or diffusion techniques), even with significantly faster and more memory-efficient training. We also show conditional and compositional generation abilities of our latent-variable EBMs as their additional benefits, even without explicit conditional training. The code is available at https://github.com/hankook/CLEL.

## 1 INTRODUCTION

Generative modeling is a fundamental machine learning task for learning complex high-dimensional data distributions $p_{\text{data}}(\mathbf{x})$. Among a number of generative frameworks, *energy-based models* (EBMs, LeCun et al., 2006; Salakhutdinov et al., 2007), whose density is proportional to the exponential negative energy, *i.e.*, $p_\theta(\mathbf{x}) \propto \exp(-E_\theta(\mathbf{x}))$, have recently gained much attention due to their attractive properties. For example, EBMs can naturally provide the explicit (unnormalized) density, unlike generative adversarial networks (GANs, Goodfellow et al., 2014). Furthermore, they are much less restrictive in architectural designs than other explicit density models such as autoregressive (Oord et al., 2016b;a) and flow-based models (Rezende & Mohamed, 2015; Dinh et al., 2017). Hence, EBMs have found wide applications, including image inpainting (Du & Mordatch, 2019), hybrid discriminative-generative models (Grathwohl et al., 2019; Yang & Ji, 2021), protein design (Ingraham et al., 2019; Du et al., 2020b), and text generation (Deng et al., 2020).

Despite the attractive properties, training EBMs has remained challenging; *e.g.*, it often suffers from the training instability due to the intractable sampling and the absence of the normalizing constant. Recently, various techniques have been developed for improving the training stability and the quality of generated samples, for example, gradient clipping (Du & Mordatch, 2019), short MCMC runs (Nijkamp et al., 2019), data augmentations in MCMC sampling (Du et al., 2021), and better divergence measures (Yu et al., 2020; 2021; Du et al., 2021). To further improve EBMs, there are several recent attempts to incorporate other generative models into EBM training, *e.g.*, variational autoencoders (VAEs) (Xiao et al., 2021), flow models (Gao et al., 2020; Xie et al., 2022), or diffusion techniques (Gao et al., 2021). However, they often require a high computational cost for training such an extra generative model, or there still exists a large gap between EBMs and state-of-the-art generative frameworks like GANs (Kang et al., 2021) or score-based models (Vahdat et al., 2021).

---

*Work done at KAIST.

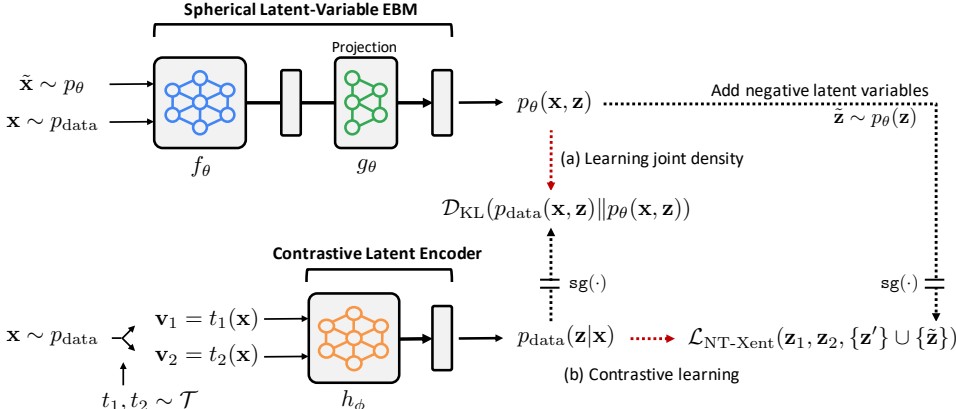

Figure 1: Illustration of the proposed Contrastive Latent-guided Energy Learning (CLEL) framework. (a) Our spherical latent-variable EBM ($f_\theta, g_\theta$) learns the joint data distribution $p_{\text{data}}(\mathbf{x}, \mathbf{z})$ generated by our contrastive latent encoder $h_\phi$. (b) The encoder $h_\phi$ is trained by contrastive learning with additional negative variables $\tilde{\mathbf{z}} \sim p_\theta(\tilde{\mathbf{z}})$. Here, $\mathbf{z}_i = h_\phi(t_i(\mathbf{x}))/\|h_\phi(t_i(\mathbf{x}))\|_2$ where $t_i \sim \mathcal{T}$ denotes a random augmentation, and $\text{sg}(\cdot)$ denotes the stop-gradient operation.

Instead of utilizing extra expensive generative models, in this paper, we ask whether EBMs can be improved by other unsupervised techniques of low cost. To this end, we are inspired by recent advances in unsupervised representation learning literature (Chen et al., 2020; Grill et al., 2020; He et al., 2021), especially by the fact that the discriminative representations can be obtained much easier than generative modeling. Interestingly, such representations have been used to detect out-of-distribution samples (Hendrycks et al., 2019a;b), so we expect that training EBMs can benefit from good representations. In particular, we primarily focus on contrastive representation learning (Oord et al., 2018; Chen et al., 2020; He et al., 2020) since it can learn instance discriminability, which has been shown to be effective in not only representation learning, but also training GANs (Jeong & Shin, 2021; Kang et al., 2021) and out-of-distribution detection (Tack et al., 2020).

In this paper, we propose *Contrastive Latent-guided Energy Learning (CLEL)*, a simple yet effective framework for improving EBMs via contrastive representation learning (CRL). Our CLEL consists of two components, which are illustrated in Figure 1.

- **Contrastive latent encoder.** Our key idea is to consider representations learned by CRL as an underlying latent variable distribution $p_{\text{data}}(\mathbf{z}|\mathbf{x})$. Specifically, we train an encoder $h_\phi$ via CRL, and treat the encoded representation $\mathbf{z} := h_\phi(\mathbf{x})$ as the true latent variable given data $\mathbf{x}$, *i.e.*, $\mathbf{z} \sim p_{\text{data}}(\cdot|\mathbf{x})$. This latent variable could guide EBMs to understand the underlying data structure more quickly and accelerate training since the latent variable contains semantic information of the data thanks to CRL. Here, we assume the latent variables are spherical, *i.e.*, $\|\mathbf{z}\|_2 = 1$, since recent CRL methods (He et al., 2020; Chen et al., 2020) use the cosine distance on the latent space.
- **Spherical latent-variable EBM.** We introduce a new class of latent-variable EBMs $p_\theta(\mathbf{x}, \mathbf{z})$ for modeling the joint distribution $p_{\text{data}}(\mathbf{x}, \mathbf{z})$ generated by the contrastive latent encoder. Since the latent variables are spherical, we separate the output vector $f := f_\theta(\mathbf{x})$ into its norm $\|f\|_2$ and direction $f/\|f\|_2$ for modeling $p_\theta(\mathbf{x})$ and $p_\theta(\mathbf{z}|\mathbf{x})$, respectively. We found that this separation technique reduces the conflict between $p_\theta(\mathbf{x})$ and $p_\theta(\mathbf{z}|\mathbf{x})$ optimizations, which makes training stable. In addition, we treat the latent variables drawn from our EBM, $\tilde{\mathbf{z}} \sim p_\theta(\mathbf{z})$, as additional negatives in CRL, which further improves our CLEL. Namely, CRL guides EBM and vice versa.[1]

We demonstrate the effectiveness of the proposed framework through extensive experiments. For example, our EBM achieves 8.61 FID under unconditional CIFAR-10 generation, which is lower than those of existing EBM models. Here, we remark that utilizing CRL into our EBM training increases training time by only 10% in our experiments (*e.g.*, 38→41 GPU hours). This enables us to achieve the lower FID score even with significantly less computational resources (*e.g.*, we use single RTX3090 GPU only) than the prior EBMs that utilize VAEs (Xiao et al., 2021) or diffusion-based recovery likelihood (Gao et al., 2021). Furthermore, even without explicit conditional training, our

---

[1]The representation quality of CRL for classification tasks is not much improved in our experiments under the joint training of CRL and EBM. Hence, we only report the performance of EBM, not that of CRL.

latent-variable EBMs naturally can provide the latent-conditional density $p_\theta(\mathbf{x}|\mathbf{z})$; we verify its effectiveness under various applications: out-of-distribution (OOD) detection, conditional sampling, and compositional sampling. For example, OOD detection using the conditional density shows superiority over various likelihood-based models. Finally, we remark that our idea is not limited to contrastive representation learning and we show EBMs can be also improved by other representation learning methods like BYOL (Grill et al., 2020) or MAE (He et al., 2021) (see Section 4.5).

## 2 PRELIMINARIES

In this work, we mainly consider unconditional generative modeling: given a set of *i.i.d.* samples $\{\mathbf{x}^{(i)}\}_{i=1}^N$ drawn from an unknown data distribution $p_{\text{data}}(\mathbf{x})$, our goal is to learn a model distribution $p_\theta(\mathbf{x})$ parameterized by $\theta$ to approximate the data distribution $p_{\text{data}}(\mathbf{x})$. To this end, we parameterize $p_\theta(\mathbf{x})$ using energy-based models (EBMs), and incorporate contrastive representation learning (CRL) into EBMs for improving them. We briefly describe the concepts of EBMs and CRL in Section 2.1 and Section 2.2, respectively, and then introduce our framework in Section 3.

### 2.1 ENERGY-BASED MODELS

An *energy-based model (EBM)* is a probability distribution on $\mathbb{R}^{d_{\mathbf{x}}}$, defined as follows: for $\mathbf{x} \in \mathbb{R}^{d_{\mathbf{x}}}$,

$$p_\theta(\mathbf{x}) = \frac{\exp(-E_\theta(\mathbf{x}))}{Z_\theta}, \qquad Z_\theta = \int_{\mathbb{R}^{d_{\mathbf{x}}}} \exp(-E_\theta(\mathbf{x}))d\mathbf{x}, \tag{1}$$

where $E_\theta(\mathbf{x})$ is the *energy function* parameterized by $\theta$ and $Z_\theta$ denotes the normalizing constant, called the *partition function*. An important application of EBMs is to find a parameter $\theta$ such that $p_\theta$ is close to $p_{\text{data}}$. A popular method for finding such $\theta$ is to minimize Kullback–Leibler (KL) divergence between $p_{\text{data}}$ and $p_\theta$ via gradient descent:

$$D_{\text{KL}}(p_{\text{data}}\|p_\theta) = -\mathbb{E}_{\mathbf{x}\sim p_{\text{data}}}[\log p_\theta(\mathbf{x})] + \text{Constant}, \tag{2}$$

$$\nabla_\theta D_{\text{KL}}(p_{\text{data}}\|p_\theta) = \mathbb{E}_{\mathbf{x}\sim p_{\text{data}}}[\nabla_\theta E_\theta(\mathbf{x})] - \mathbb{E}_{\tilde{\mathbf{x}}\sim p_\theta}[\nabla_\theta E_\theta(\tilde{\mathbf{x}})]. \tag{3}$$

Since this gradient computation (3) is NP-hard in general (Jerrum & Sinclair, 1993), it is often approximated via Markov chain Monte Carlo (MCMC) methods. In this work, we use the stochastic gradient Langevin dynamics (SGLD, Welling & Teh, 2011), a gradient-based MCMC method for approximate sampling. Specifically, at the $(t+1)$-th iteration, SGLD updates the current sample $\tilde{\mathbf{x}}^{(t)}$ to $\tilde{\mathbf{x}}^{(t+1)}$ using the following procedure:

$$\tilde{\mathbf{x}}^{(t+1)} \leftarrow \tilde{\mathbf{x}}^{(t)} + \frac{\varepsilon^2}{2}\underbrace{\nabla_{\mathbf{x}}\log p_\theta(\tilde{\mathbf{x}}^{(t)})}_{=-\nabla_{\mathbf{x}}E_\theta(\tilde{\mathbf{x}}^{(t)})} + \varepsilon\boldsymbol{\delta}^{(t)}, \quad \boldsymbol{\delta}^{(t)} \sim \mathcal{N}(0, I), \tag{4}$$

where $\varepsilon > 0$ is some predefined constant, $\tilde{\mathbf{x}}^{(0)}$ denotes an initial state, and $\mathcal{N}$ denotes the multivariate normal distribution. Here, it is known that the distribution of $\tilde{\mathbf{x}}^{(T)}$ (weakly) converges to $p_\theta$ with small enough $\varepsilon$ and large enough $T$ under various assumptions (Vollmer et al., 2016; Raginsky et al., 2017; Xu et al., 2018; Zou et al., 2021).

**Latent-variable energy-based models.** EBMs can naturally incorporate a latent variable by specifying the joint density $p_\theta(\mathbf{x}, \mathbf{z}) \propto \exp(-E_\theta(\mathbf{x}, \mathbf{z}))$ of observed data $\mathbf{x}$ and the latent variable $\mathbf{z}$. This class includes a number of EBMs: *e.g.*, deep Boltzmann machines (Salakhutdinov & Hinton, 2009) and conjugate EBMs (Wu et al., 2021). Similar to standard EBMs, these latent-variable EBMs can be trained by minimizing KL divergence between $p_{\text{data}}(\mathbf{x})$ and $p_\theta(\mathbf{x})$ as described in (3):

$$\nabla_\theta D_{\text{KL}}(p_{\text{data}}\|p_\theta) = \mathbb{E}_{\mathbf{x}\sim p_{\text{data}}(\mathbf{x})}[\nabla_\theta E_\theta(\mathbf{x})] - \mathbb{E}_{\tilde{\mathbf{x}}\sim p_\theta(\tilde{\mathbf{x}})}[\nabla_\theta E_\theta(\tilde{\mathbf{x}})] \tag{5}$$

$$= \mathbb{E}_{\mathbf{x}\sim p_{\text{data}}(\mathbf{x}),\mathbf{z}\sim p_\theta(\mathbf{z}|\mathbf{x})}[\nabla_\theta E_\theta(\mathbf{x}, \mathbf{z})] - \mathbb{E}_{\tilde{\mathbf{x}}\sim p_\theta(\tilde{\mathbf{x}}),\mathbf{z}\sim p_\theta(\mathbf{z}|\tilde{\mathbf{x}})}[\nabla_\theta E_\theta(\tilde{\mathbf{x}}, \mathbf{z})], \tag{6}$$

where $E_\theta(\mathbf{x})$ is the marginal energy, *i.e.*, $E_\theta(\mathbf{x}) = -\log \int \exp(-E_\theta(\mathbf{x}, \mathbf{z}))d\mathbf{z}$.

### 2.2 CONTRASTIVE REPRESENTATION LEARNING

Generally speaking, *contrastive learning* aims to learn a meaningful representation by minimizing distance between similar (*i.e.*, positive) samples, and maximizing distance between dissimilar (*i.e.*,

negative) samples on the representation space. Formally, let $h_\phi : \mathbb{R}^{d_\mathbf{x}} \to \mathbb{R}^{d_\mathbf{z}}$ be a $\phi$-parameterized encoder, $(\mathbf{x}, \mathbf{x}_+)$ and $(\mathbf{x}, \mathbf{x}_-)$ be positive and negative pairs, respectively. Contrastive learning then maximizes $\text{sim}(h_\phi(\mathbf{x}), h_\phi(\mathbf{x}_+))$ and minimizes $\text{sim}(h_\phi(\mathbf{x}), h_\phi(\mathbf{x}_-))$ where $\text{sim}(\cdot, \cdot)$ is a similarity metric defined on the representation space $\mathbb{R}^{d_\mathbf{z}}$.

Under the unsupervised setup, various methods for constructing positive and negative pairs have been proposed: *e.g.*, data augmentations (He et al., 2020; Chen et al., 2020; Tian et al., 2020), spatial or temporal co-occurrence (Oord et al., 2018), and image channels (Tian et al., 2019). In this work, we mainly focus on a popular contrastive learning framework, SimCLR (Chen et al., 2020), which constructs positive and negative pairs via various data augmentations such as cropping and color jittering. Specifically, given a mini-batch $\mathcal{B} = \{\mathbf{x}^{(i)}\}_{i=1}^n$, SimCLR first constructs two augmented views $\{\mathbf{v}_j^{(i)} := t_j^{(i)}(\mathbf{x}^{(i)})\}_{j \in \{1,2\}}$ for each data sample $\mathbf{x}^{(i)}$ via random augmentations $t_j^{(i)} \sim \mathcal{T}$. Then, it considers $(\mathbf{v}_1^{(i)}, \mathbf{v}_2^{(i)})$ as a positive pair and $(\mathbf{v}_1^{(i)}, \mathbf{v}_2^{(k)})$ as a negative pair for all $k \neq i$. The SimCLR objective $\mathcal{L}_{\text{SimCLR}}$ is defined as follows:

$$\mathcal{L}_{\text{SimCLR}}(\mathcal{B}; \phi, \tau) = \frac{1}{2n} \sum_{i=1}^n \sum_{j=1,2} \mathcal{L}_{\text{NT-Xent}} \left( h_\phi(\mathbf{v}_j^{(i)}), h_\phi(\mathbf{v}_{3-j}^{(i)}), \{h_\phi(\mathbf{v}_l^{(k)})\}_{k \neq i, l \in \{1,2\}}; \tau \right), \quad (7)$$

$$\mathcal{L}_{\text{NT-Xent}}(\mathbf{z}, \mathbf{z}_+, \{\mathbf{z}_-\}; \tau) = -\log \frac{\exp(\text{sim}(\mathbf{z}, \mathbf{z}_+)/\tau)}{\exp(\text{sim}(\mathbf{z}, \mathbf{z}_+)/\tau) + \sum_{\mathbf{z}_-} \exp(\text{sim}(\mathbf{z}, \mathbf{z}_-)/\tau)}, \quad (8)$$

where $\text{sim}(\mathbf{u}, \mathbf{v}) = \mathbf{u}^\top \mathbf{v} / \|\mathbf{u}\|_2 \|\mathbf{v}\|_2$ is the cosine similarity, $\tau$ is a hyperparameter for temperature scaling, and $\mathcal{L}_{\text{NT-Xent}}$ denotes the normalized temperature-scaled cross entropy (Chen et al., 2020).

## 3 METHOD

Recall that our goal is to learn an energy-based model (EBM) $p_\theta(\mathbf{x}) \propto \exp(-E_\theta(\mathbf{x}))$ to approximate a complex underlying data distribution $p_{\text{data}}(\mathbf{x})$. In this work, we propose Contrastive Latent-guided Energy Learning (CLEL), a simple yet effective framework for improving EBMs via contrastive representation learning. Our key idea is that directly incorporating with *semantically meaningful contexts* of data could improve EBMs. To this end, we consider the (random) representation $\mathbf{z} \sim p_{\text{data}}(\mathbf{z}|\mathbf{x})$ of $\mathbf{x}$, generated by contrastive learning, as the underlying latent variable.[2] Namely, we model the joint distribution $p_{\text{data}}(\mathbf{x}, \mathbf{z}) = p_{\text{data}}(\mathbf{x})p_{\text{data}}(\mathbf{z}|\mathbf{x})$ via a latent-variable EBM $p_\theta(\mathbf{x}, \mathbf{z})$. Our intuition on the benefit of modeling $p_{\text{data}}(\mathbf{x}, \mathbf{z})$ is two-fold: (i) conditional generative modeling $p_{\text{data}}(\mathbf{x}|\mathbf{z})$ given some good contexts (e.g., labels) of data is much easier than unconditional modeling $p_{\text{data}}(\mathbf{x})$ (Mirza & Osindero, 2014; Van den Oord et al., 2016; Reed et al., 2016), and (ii) the mode collapse problem of generation can be resolved by predicting the contexts $p_{\text{data}}(\mathbf{z}|\mathbf{x})$ (Odena et al., 2017; Bang & Shim, 2021). The detailed implementations of $p_{\text{data}}(\mathbf{z}|\mathbf{x})$, called the contrastive latent encoder, and the latent-variable EBM are described in Section 3.1 and 3.2, respectively, while Section 3.3 presents how to train them in detail. Our overall framework is illustrated in Figure 1.

### 3.1 CONTRASTIVE LATENT ENCODER

To construct a meaningful latent distribution $p_{\text{data}}(\mathbf{z}|\mathbf{x})$ for improving EBMs, we use contrastive representation learning. To be specific, we first train a latent encoder $h_\phi : \mathbb{R}^{d_\mathbf{x}} \to \mathbb{R}^{d_\mathbf{z}}$, which is a deep neural network (DNN) parameterized by $\phi$, using a variant of the SimCLR objective $\mathcal{L}_{\text{SimCLR}}$ (7) (we describe its detail in Section 3.3) with a random augmentation distribution $\mathcal{T}$. Since our objective only measures the cosine similarity between distinct representations, one can consider the encoder $h_\phi$ maps a randomly augmented sample to a *unit vector*. We define the latent sampling procedure $\mathbf{z} \sim p_{\text{data}}(\mathbf{z}|\mathbf{x})$ as follows:

$$\mathbf{z} \sim p_{\text{data}}(\mathbf{z}|\mathbf{x}) \quad \Leftrightarrow \quad \mathbf{z} = h_\phi(t(\mathbf{x}))/\|h_\phi(t(\mathbf{x}))\|_2, t \sim \mathcal{T}. \quad (9)$$

### 3.2 SPHERICAL LATENT-VARIABLE ENERGY-BASED MODELS

We use a DNN $f_\theta : \mathbb{R}^{d_\mathbf{x}} \to \mathbb{R}^{d_\mathbf{z}}$ parameterized by $\theta$ for modeling $p_\theta(\mathbf{x}, \mathbf{z})$. Following that the latent variable $\mathbf{z} \sim p_{\text{data}}(\mathbf{z}|\mathbf{x})$ is on the unit sphere, we utilize the directional information $f_\theta(\mathbf{x})/\|f_\theta(\mathbf{x})\|_2$

---

[2]Chen et al. (2020) shows that the contrastive representations contains such contexts under various tasks.

for modeling $p_\theta(\mathbf{z}|\mathbf{x})$, while the remaining information $\|f_\theta(\mathbf{x})\|_2$ is used for modeling $p_\theta(\mathbf{x})$. We empirically found that this norm-direction separation stabilizes the latent-variable EBM training.[3] For better modeling $p_\theta(\mathbf{z}|\mathbf{x})$, we additionally apply a directional projector $g_\theta : \mathbb{S}^{d_\mathbf{z}-1} \to \mathbb{S}^{d_\mathbf{z}-1}$ to $f_\theta(\mathbf{x})/\|f_\theta(\mathbf{x})\|_2$, which is constructed by a two-layer MLP, followed by $\ell_2$ normalization. We found that it is useful for narrowing the gap between distributions of the direction $f_\theta(\mathbf{x})/\|f_\theta(\mathbf{x})\|_2$ and the *uniformly-distributed* latent variable $p_{\text{data}}(\mathbf{z})$ (see Section 4.5 and Appendix E for detailed discussion). Overall, we define the joint energy $E_\theta(\mathbf{x}, \mathbf{z})$ as follows:

$$E_\theta(\mathbf{x}, \mathbf{z}) = \frac{1}{2}\|f_\theta(\mathbf{x})\|_2^2 - \beta g_\theta \left( \frac{f_\theta(\mathbf{x})}{\|f_\theta(\mathbf{x})\|_2} \right)^\top \mathbf{z}, \tag{10}$$

$$E_\theta(\mathbf{x}) = -\log \int_{\mathbb{S}^{d-1}} \exp(-E_\theta(\mathbf{x}, \mathbf{z}))d\mathbf{z} = \frac{1}{2}\|f_\theta(\mathbf{x})\|_2^2 + \text{Constant}, \tag{11}$$

where $\beta \geq 0$ is a hyperparameter. Note that the marginal energy $E_\theta(\mathbf{x})$ only depends on $\|f_\theta(\mathbf{x})\|_2$ since $\int_{\mathbb{S}^{d-1}} \exp(\beta g_\theta (f_\theta(\mathbf{x})/\|f_\theta(\mathbf{x})\|_2)^\top \mathbf{z})d\mathbf{z}$ is independent of $\mathbf{x}$ due to the symmetry. Also, the norm-based design does not sacrifice the flexibility for energy modeling (see Appendix F for details).

## 3.3 Training

Remark that Section 3.1 and 3.2 define $p_{\text{data}}(\mathbf{x}, \mathbf{z})$ and $p_\theta(\mathbf{x}, \mathbf{z})$, respectively. We now describe how to train the contrastive latent encoder $h_\phi$ and the spherical latent-variable EBM $p_\theta$ via mini-batch stochastic optimization algorithms in detail (see Appendix A for the pseudo-code).

Let $\{\mathbf{x}^{(i)}\}_{i=1}^n$ be real samples randomly drawn from the training dataset. We first generate $n$ samples $\{\tilde{\mathbf{x}}^{(i)}\}_{i=1}^n \sim p_\theta(\mathbf{x})$ using the current EBM via stochastic gradient Langevin dynamics (SGLD) (4). Here, to reduce the computational complexity and improve the generation quality of SGLD, we use two techniques: a replay buffer to maintain Markov chains persistently (Du & Mordatch, 2019), and periodic data augmentation transitions to encourage exploration (Du et al., 2021). We then draw latent variables from $p_{\text{data}}$ and $p_\theta$: $\mathbf{z}^{(i)} \sim p_{\text{data}}(\mathbf{z}|\mathbf{x}^{(i)})$ and $\tilde{\mathbf{z}}^{(i)} \sim p_\theta(\mathbf{z}|\tilde{\mathbf{x}}^{(i)})$ for all $i$. For the latter case, we simply use the mode of $p_\theta(\mathbf{z}|\mathbf{x}^{(i)})$ instead of sampling, namely, $\tilde{\mathbf{z}}^{(i)} := g_\theta(f_\theta(\mathbf{x})/\|f_\theta(\mathbf{x})\|_2)$. Let $\mathcal{B} := \{(\mathbf{x}^{(i)}, \mathbf{z}^{(i)})\}_{i=1}^n$ and $\tilde{\mathcal{B}} := \{(\tilde{\mathbf{x}}^{(i)}, \tilde{\mathbf{z}}^{(i)})\}_{i=1}^n$ be real and generated mini-batches, respectively.

Under this setup, we define the objective $\mathcal{L}_{\text{EBM}}$ for the EBM parameter $\theta$ as follows:

$$\mathcal{L}_{\text{EBM}}(\mathcal{B}, \tilde{\mathcal{B}}; \theta, \alpha, \beta) = \frac{1}{n}\sum_{i=1}^n E_\theta(\mathbf{x}^{(i)}, \mathbf{z}^{(i)}) - E_\theta(\tilde{\mathbf{x}}^{(i)}) + \alpha \cdot (E_\theta(\mathbf{x}^{(i)})^2 + E_\theta(\tilde{\mathbf{x}}^{(i)})^2), \tag{12}$$

where the first two terms correspond to the empirical average of $D_{\text{KL}}(p_{\text{data}}\|p_\theta)$[4] and $\alpha$ is a hyperparameter for energy regularization to prevent divergence, following Du & Mordatch (2019). When training the latent encoder $h_\phi$ via contrastive learning, we use the SimCLR (Chen et al., 2020) loss $\mathcal{L}_{\text{SimCLR}}$ (7) with additional negative latent variables $\{\tilde{\mathbf{z}}^{(i)}\}_{i=1}^n$. To be specific, we define the objective $\mathcal{L}_{\text{LE}}$ for the latent encoder parameter $\phi$ as follows:

$$\mathcal{L}_{\text{LE}}(\mathcal{B}, \tilde{\mathcal{B}}; \phi, \tau) = \frac{1}{2n}\sum_{i=1}^n \sum_{j=1,2} \mathcal{L}_{\text{NT-Xent}} \left( \mathbf{z}_j^{(i)}, \mathbf{z}_{3-j}^{(i)}, \{\mathbf{z}_l^{(k)}\}_{k \neq i, l \in \{1,2\}} \cup \{\tilde{\mathbf{z}}^{(i)}\}_{i=1}^n; \tau \right), \tag{13}$$

where $\mathcal{L}_{\text{NT-Xent}}$ is the normalized temperature-scaled cross entropy defined in (8), $\tau$ is a hyperparameter for temperature scaling, $\mathbf{z}_j^{(i)} := h_\phi(t_j^{(i)}(\mathbf{x}^{(i)}))$, and $\{t_j^{(i)}\} \sim \mathcal{T}$ are random augmentations. We found that considering $\{\tilde{\mathbf{z}}^{(i)}\}_{i=1}^n$ as negative representations for contrastive learning increases the latent diversity, which further improves the generation quality in our CLEL framework.

To sum up, our CLEL jointly optimizes the latent encoder $h_\phi$ and the latent-variable EBM $(f_\theta, g_\theta)$ from scratch via the following optimization: $\min_{\phi,\theta} \mathbb{E}_{\mathcal{B}, \tilde{\mathcal{B}}}[\mathcal{L}_{\text{EBM}}(\mathcal{B}, \tilde{\mathcal{B}}; \theta, \alpha, \beta) + \mathcal{L}_{\text{LE}}(\mathcal{B}, \tilde{\mathcal{B}}; \phi, \tau)]$. After training, we only utilize our latent-variable EBM $(f_\theta, g_\theta)$ when generating samples. The latent encoder $h_\phi$ is used only when extracting a representation of a specific sample during training.

---

[3]For example, we found a multi-head architecture for modeling $p_\theta(\mathbf{x})$ and $p_\theta(\mathbf{z}|\mathbf{x})$ makes training unstable. We provide detailed discussion and supporting experiments in Appendix D.

[4]Here, $\tilde{\mathbf{z}}$ is unnecessary since $\mathbb{E}_{\tilde{\mathbf{z}} \sim p_\theta(\mathbf{z}|\tilde{\mathbf{x}})}[\nabla_\theta E_\theta(\tilde{\mathbf{x}}, \tilde{\mathbf{z}})] = \nabla_\theta E_\theta(\tilde{\mathbf{x}})$.

Table 1: FID scores for unconditional generation on CIFAR-10 (Krizhevsky et al., 2009). † denotes EBMs that utilize auxiliary generators, and ‡ denotes hybrid discriminative-generative models.

| Method | FID | Method | FID |
|---|---|---|---|
| *Energy-based models (EBMs)* | | *Other likelihood models* | |
| Short-run EBM (Nijkamp et al., 2019) | 44.50 | PixelCNN (Oord et al., 2016b) | 65.93 |
| JEM‡ (Grathwohl et al., 2019) | 38.40 | NVAE (Vahdat & Kautz, 2020) | 51.67 |
| IGEBM (Du & Mordatch, 2019) | 38.20 | Glow (Kingma & Dhariwal, 2018) | 48.90 |
| FlowCE† (Gao et al., 2020) | 37.30 | NCP-VAE (Aneja et al., 2021) | 24.08 |
| VERA†‡ (Grathwohl et al., 2021) | 27.50 | *Score-based models* | |
| Improved CD (Du et al., 2021) | 25.10 | | |
| BiDVL (Kan et al., 2022) | 20.75 | NCSN (Song & Ermon, 2019) | 25.30 |
| GEBM† (Arbel et al., 2021) | 19.31 | NCSNv2 (Song & Ermon, 2020) | 10.87 |
| CF-EBM (Zhao et al., 2021) | 16.71 | DDPM (Ho et al., 2020) | 3.17 |
| CoopFlow† (Xie et al., 2022) | 15.80 | NCSN++ (Song et al., 2021) | 2.20 |
| **CLEL-Base (Ours)** | 15.27 | *GAN-based models* | |
| VAEBM† (Xiao et al., 2021) | 12.19 | | |
| EBM-Diffusion (Gao et al., 2021) | 9.58 | StyleGAN2-DiffAugment (Zhao et al., 2020) | 5.79 |
| **CLEL-Large (Ours)** | **8.61** | StyleGAN2-ADA (Karras et al., 2020) | 2.92 |

Table 2: FID scores for unconditional generation on ImageNet 32×32.

| Method | FID |
|---|---|
| IGEBM | 62.23 |
| PixelCNN | 40.51 |
| Improved CD | 32.48 |
| CF-EBM | 26.31 |
| **CLEL-Base (Ours)** | **22.16** |
| **CLEL-Large (Ours)** | **15.47** |

Table 3: FID improvements via different configurations with training time and GPU memory footprint on single RTX3090 GPU of 24G memory. Underline is based on our estimation as the model cannot be trained on the single GPU.

| Method | Params (M) | Time | Memory | FID |
|---|---|---|---|---|
| Baseline w/o CLEL | 6.96 | 38h | 6G | 23.50 |
| + CLEL (**Base**) | 6.96 | 41h | 7G | 15.27 |
| + multi-scale architecture | 19.29 | 74h | 8G | 12.46 |
| + CRL with a batch size of 256 | 19.29 | 76h | 10G | 11.65 |
| + more channels (**Large**) | 30.70 | 133h | 11G | **8.61** |
| EBM-Diffusion ($N = 2$ blocks) | 9.06 | 163h | 10G | 17.34 |
| EBM-Diffusion ($N = 8$ blocks) | 34.83 | 652h | 40G | 9.58 |
| VAEBM | 135.88 | 414h | 129G | 12.19 |

## 4 EXPERIMENTS

We verify the effectiveness of our Contrastive Latent-guided Energy Learning (CLEL) framework under various scenarios: (a) unconditional generation (Section 4.1), (b) out-of-distribution detection (Section 4.2), (c) conditional sampling (Section 4.3), and (d) compositional sampling (Section 4.4). All the architecture, training, and evaluation details are described in Appendidx B.

### 4.1 UNCONDITIONAL IMAGE GENERATION

An important application of EBMs is to generate images using the energy function $E_\theta(\mathbf{x})$. To this end, we train our CLEL framework on CIFAR-10 (Krizhevsky et al., 2009) and ImageNet 32×32 (Deng et al., 2009; Chrabaszcz et al., 2017) under the unsupervised setting. We then generate 50k samples using SGLD and evaluate their qualities using Fréchet Inception Distance (FID) scores (Heusel et al., 2017; Seitzer, 2020). The unconditionally generated samples are provided in Figure 2.

Table 1 and 2 show the FID scores of our CLEL and other generative models for unconditional generation on CIFAR-10 and ImageNet 32×32, respectively. We first find that CLEL outperforms previous EBMs under both CIFAR-10 and ImageNet 32×32 datasets. As shown in Table 3, our method can benefit from a multi-scale architecture as Du et al. (2021) did, contrastive representation learning (CRL) with a larger batch, more channels at lower layers in our EBM $f_\theta$. As a result, we achieve 8.61 FID on CIFAR-10, which is lower than that of the prior-art EBM based on diffusion recovery likelihood, EBM-Diffusion (Gao et al., 2021), even with 5× faster and 4× more memory-efficient training (when using the similar number of parameters for EBMs). Then, we narrow the gap between EBMs and state-of-the-art frameworks like GANs without help from other generative models. We think our CLEL can be further improved by incorporating an auxiliary generator (Arbel et al., 2021; Xiao et al., 2021) or diffusion (Gao et al., 2021), and we leave it for future work.

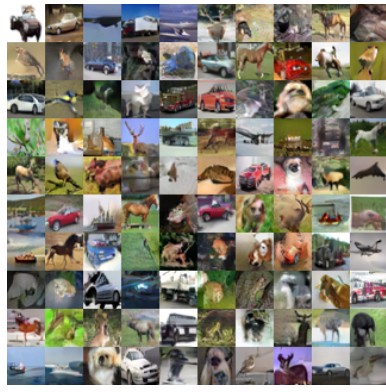
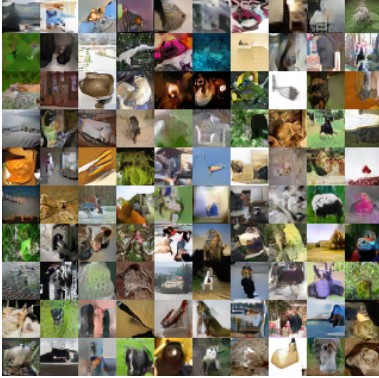

(a) CIFAR-10                              (b) ImageNet 32×32

Figure 2: Unconditional generated samples from our EBMs on CIFAR-10 (Krizhevsky et al., 2009) and ImageNet 32×32 (Deng et al., 2009; Chrabaszcz et al., 2017).

Table 4: AUROC scores in OOD detection using explicit density models on CIFAR-10. **Bold** and underline entries indicates the best and second best, respectively, among unsupervised methods, where JEM and VERA are supervised methods.

| Method | SVHN | Textures | CIFAR10 Interp. | CIFAR100 | CelebA |
|---|---|---|---|---|---|
| PixelCNN++ (Salimans et al., 2017) | 0.32 | 0.33 | 0.71 | 0.63 | - |
| GLOW (Kingma & Dhariwal, 2018) | 0.24 | 0.27 | 0.51 | 0.55 | 0.57 |
| NVAE (Vahdat & Kautz, 2020) | 0.42 | - | 0.64 | 0.56 | 0.68 |
| IGEBM (Du & Mordatch, 2019) | 0.63 | 0.48 | 0.70 | 0.50 | 0.70 |
| VAEBM (Xiao et al., 2021) | 0.83 | - | 0.70 | 0.62 | 0.77 |
| Improved CD (Du et al., 2021) | 0.91 | 0.88 | 0.65 | **0.83** | - |
| **CLEL-Base (Ours)** | **0.9848** | **0.9437** | **0.7248** | 0.7161 | **0.7717** |
| JEM (Grathwohl et al., 2019) | 0.67 | 0.60 | 0.65 | 0.67 | 0.75 |
| VERA (Grathwohl et al., 2021). | 0.83 | - | 0.86 | 0.73 | 0.33 |

## 4.2 OUT-OF-DISTRIBUTION DETECTION

EBMs can be also used for detecting out-of-distribution (OOD) samples. For the OOD sample detection, previous EBM-based approaches often use the (marginal) unnormalized likelihood $p_\theta(\mathbf{x}) \propto \exp(-E_\theta(\mathbf{x}))$. In contrast, our CLEL is capable of modeling the joint density $p_\theta(\mathbf{x}, \mathbf{z}) \propto E_\theta(\mathbf{x}, \mathbf{z})$. Using this capability, we propose an energy-based OOD detection score: given $\mathbf{x}$,

$$s(\mathbf{x}) := \frac{1}{2}\|f_\theta(\mathbf{x})\|_2^2 - \beta g_\theta \left( \frac{f_\theta(\mathbf{x})}{\|f_\theta(\mathbf{x})\|_2} \right)^\top \frac{h_\phi(\mathbf{x})}{\|h_\phi(\mathbf{x})\|_2}. \tag{14}$$

We found that the second term in (14) helps to detect the semantic difference between in- and out-of-distribution samples. Table 4 shows our CLEL's superiority over other explicit density models in OOD detection, especially when OOD samples are drawn from different domains, *e.g.*, SVHN (Netzer et al., 2011) and Texture (Cimpoi et al., 2014) datasets.

## 4.3 CONDITIONAL SAMPLING

One advantage of latent-variable EBMs is that they can offer the latent-conditional density $p_\theta(\mathbf{x}|\mathbf{z}) \propto \exp(-E_\theta(\mathbf{x}, \mathbf{z}))$. Hence, our EBMs can enjoy the advantage even though CLEL does not explicitly train conditional models. To verify this, we first test *instance-conditional sampling*: given a real sample $\mathbf{x}$, we draw the underlying latent variable $\mathbf{z} \sim p_{\text{data}}(\mathbf{z}|\mathbf{x})$ using our latent encoder $h_\phi$, and then perform SGLD sampling using our joint energy $E_\theta(\mathbf{x}, \mathbf{z})$ defined in (10). We here use our CIFAR-10 model. As shown in Figure 3a, the instance-conditionally generated samples contain similar information (*e.g.*, color, shape, and background) to the given instance.

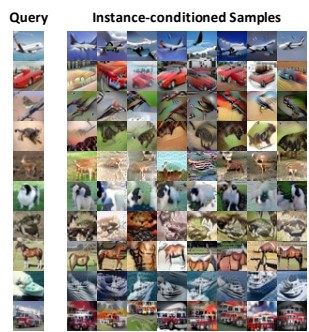 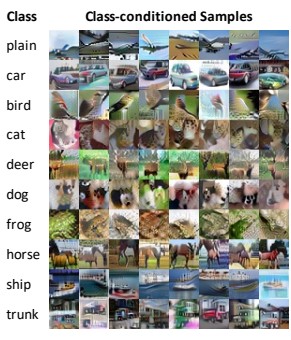 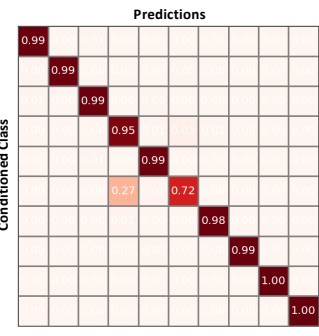

(a) Instance-conditional sampling   (b) Class-conditional sampling   (c) Confusion matrix

Figure 3: (a, b) Instance- and class-conditionally generated samples using our CLEL in CIFAR-10. (c) Confusion matrix for the class-conditionally generated samples computed by an external classifier.

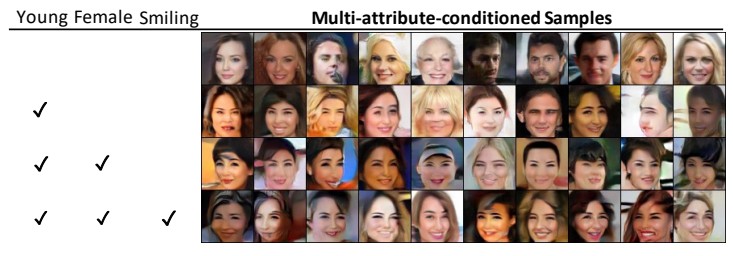 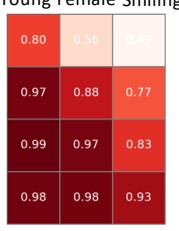

(a) Multi-attribute-conditional sampling   (b) Attribute predictions

Figure 4: Compositional generation results in CelebA. (a) Samples are generated by conditioning on checked attributes. (b) Attribute predictions of generated samples computed by an external classifier.

This successful result motivates us to extend the sampling procedure: given a set of instances $\{\mathbf{x}^{(i)}\}$, can we generate samples that contain the shared information in $\{\mathbf{x}^{(i)}\}$? To this end, we first draw latent variables $\mathbf{z}^{(i)} \sim p_{\text{data}}(\cdot|\mathbf{x}^{(i)})$ for all $i$, and then aggregate them by summation and normalization: $\bar{\mathbf{z}} := \sum_i \mathbf{z}^{(i)}/\|\sum_i \mathbf{z}^{(i)}\|_2$. To demonstrate that samples generated from $p_\theta(\mathbf{x}|\bar{\mathbf{z}})$ contains the shared information in $\{\mathbf{x}^{(i)}\}$, we collect the set of instances $\{\mathbf{x}_y^{(i)}\}$ for each label $y$ in CIFAR-10, and check whether $\tilde{\mathbf{x}}_y \sim p_\theta(\cdot|\bar{\mathbf{z}}_y)$ has the same label $y$. Figure 3b shows the class-conditionally generated samples $\{\tilde{\mathbf{x}}_y\}$ and Figure 3c presents the confusion matrix of predictions for $\{\tilde{\mathbf{x}}_y\}$ computed by an external classifier $c$. Formally, each $(i, j)$-th entry is equal to $\mathbb{P}_{\tilde{\mathbf{x}}_i}(c(\tilde{\mathbf{x}}_i) = j)$. We found that $\tilde{\mathbf{x}}_y$ is likely to be predicted as the label $y$, except the case when $y$ is dog: the generated dog images sometimes look like a semantically similar class, cat. These results verify that our EBM can generate samples conditioning on a instance or class label, even without explicit conditional training.

## 4.4 COMPOSITIONALITY VIA LATENT VARIABLES

An intriguing property of EBMs is compositionality (Du et al., 2020a): given two EBMs $E(\mathbf{x}|c_1)$ and $E(\mathbf{x}|c_2)$ that are conditional energies on concepts $c_1$ and $c_2$, respectively, one can construct a new energy conditioning on both concepts: $p_\theta(\mathbf{x}|c_1 \text{ and } c_2) \propto \exp(-E(\mathbf{x}|c_1) - E(\mathbf{x}|c_2))$. As shown in Section 4.3, our CLEL implicitly learns $E(\mathbf{x}|\mathbf{z})$, and a latent variable $\mathbf{z}$ can be considered as a concept, *e.g.*, instance or class. Hence, in this section, we test compositionality of our model. To this end, we additionally train our CLEL in CelebA $64 \times 64$ (Liu et al., 2015). For compositional sampling, we first acquire three attribute vectors $\bar{\mathbf{z}}_a$ for $a \in \mathcal{A} := \{\text{Young}, \text{Female}, \text{Smiling}\}$ as we did in Section 4.3, then generate samples from a composition of conditional energies as follows:

$$E_\theta(\mathbf{x}|\mathcal{A}) := \frac{1}{2}\|f_\theta(\mathbf{x})\|_2 - \beta \sum_{a \in \mathcal{A}} \text{sim}(g_\theta(f_\theta(\mathbf{x})/\|f_\theta(\mathbf{x})\|_2), \bar{\mathbf{z}}_a), \tag{15}$$

where $\text{sim}(\cdot, \cdot)$ is the cosine similarity. Figure 4a and 4b show the generated samples conditioning on multiple attributes and their attribute prediction results computed by an external classifier, respectively. They verify our compositionality qualitatively and quantitatively. For example, almost generated faces conditioned by $\{\text{Young}, \text{Female}\}$ look young and female (see the third row in Figure 4.)

Table 5: Component ablation experiments.

| | Projection $g_\theta$ | Negative $p_\theta(\tilde{\mathbf{z}})$ | FID↓ | OOD↑ |
|---|---|---|---|---|
| (a) | Baseline ($\beta = 0$) | | 42.46 | 0.8532 |
| (b) | MLP | | 36.29 | 0.8580 |
| (c) | MLP | ✓ | **35.73** | **0.8723** |
| (d) | Identity | ✓ | 86.02 | 0.8474 |
| (e) | Linear | ✓ | 37.35 | 0.8540 |

Table 6: $\beta$ sensitivity.

| $\beta$ | FID↓ | OOD↑ |
|---|---|---|
| 0 | 42.46 | 0.8532 |
| 0.001 | 37.44 | 0.8485 |
| 0.01 | **35.73** | **0.8723** |
| 0.1 | 56.39 | 0.7559 |

Table 7: Compatibility.

| SSRL | FID↓ | OOD↑ |
|---|---|---|
| | 42.46 | 0.8532 |
| SimCLR | **35.73** | 0.8723 |
| BYOL | 36.31 | **0.8792** |
| MAE | 37.67 | 0.8561 |

## 4.5 ABLATION STUDY

**Component analysis.** To verify the importance of our CLEL's components, we conduct ablation experiments with training a smaller ResNet (He et al., 2016) in CIFAR-10 (Krizhevsky et al., 2009) for 50k training iterations. Then, we evaluate the quality of energy functions using FID and OOD detection scores. Here, we use SVHN (Netzer et al., 2011) as the OOD dataset. Table 5 demonstrates the effectiveness of CLEL's components. First, we observe that learning $p_\theta(\mathbf{z}|\mathbf{x})$ to approximate $p_{\text{data}}(\mathbf{z}|\mathbf{x})$ plays a crucial role for improving generation (see (a) *vs.* (b)). In addition, using generated latent variables $\tilde{\mathbf{z}} \sim p_\theta(\cdot)$ as negatives for contrastive learning further improves not only generation, but also OOD detection performance (see (b) *vs.* (c)). We also empirically found that using an additional projection head is critical; without projection $g_\theta$ (*i.e.*, (d)), our EBM failed to approximate $p_{\text{data}}(\mathbf{x})$, but an additional projection head (*i.e.*, (c) or (e)) makes learning feasible. Hence, we use a 2-layer MLP (c) in all experiments since it is better than a simple linear function (e). We also test various $\beta \in \{0.1, 0.01, 0.001\}$ under this evaluation setup (see Table 6) and find $\beta = 0.01$ is the best.

**Compatibility with other self-supervised representation learning methods.** While we have mainly focused on utilizing contrastive representation learning (CRL), our framework CLEL is not limited to CRL for learning the latent encoder $h_\phi$. To verify this compatibility, we replace SimCLR with other self-supervised representation learning (SSRL) methods, BYOL (Grill et al., 2020) and MAE (He et al., 2021). See Appendix C for implementation details. Note that these methods have several advantages compared to SimCLR: e.g., BYOL does not require negative pairs, and MAE does not require heavy data augmentations. Table 7 implies that any SSRL methods can be used to improve EBMs under our framework, where the CRL method, SimCLR (Chen et al., 2020), is the best.

## 5 RELATED WORKS

Energy-based models (EBMs) can offer an explicit density and are less restrictive in architecture design, but training them has been challenging. For example, it often suffers from the training instability due to the time-consuming and unstable MCMC sampling procedure (*e.g.*, a large number of SGLD steps). To reduce the computational complexity and improve the quality of generated samples, various techniques have been proposed: a replay buffer (Du & Mordatch, 2019), short-run MCMC (Nijkamp et al., 2019), augmentation-based MCMC transitions (Du et al., 2021). Recently, researchers have also attempted to incorporate other generative frameworks into EBM training *e.g.*, adversarial training (Kumar et al., 2019; Arbel et al., 2021; Grathwohl et al., 2021), flow-based models (Gao et al., 2020; Nijkamp et al., 2022; Xie et al., 2022), variational autoencoders (Xiao et al., 2021), and diffusion techniques (Gao et al., 2021). Another direction is on developing better divergence measures, *e.g.*, $f$-divergence (Yu et al., 2020), pseudo-spherical scoring rule (Yu et al., 2021), and improved contrastive divergence (Du et al., 2021). Compared to the recent advances in the EBM literature, we have focused on an orthogonal research direction that investigates how to incorporate discriminative representations, especially of contrastive learning, into training EBMs.

## 6 CONCLUSION

The early advances in deep learning was initiated from the pioneering energy-based model (EBM) works, e.g., restricted and deep Boltzman machines (Salakhutdinov et al., 2007; Salakhutdinov & Hinton, 2009), however the recent accomplishments rather rely on other generative frameworks such as diffusion models (Ho et al., 2020; Song et al., 2021). To narrow the gap, in this paper, we suggest to utilize discriminative representations for improving EBMs, and achieve significant improvements. We hope that our work would shed light again on the potential of EBMs, and would guide many further research directions for EBMs.

## ETHICS STATEMENT

In recent years, generative models have been successful in synthesizing diverse high-fidelity images. However, high-quality image generation techniques have threats to be abused for unethical purposes, e.g., creating sexual photos of an individual. This significant threats call us researchers for the future efforts on developing techniques to detect such misuses. Namely, we should learn the data distribution $p(\mathbf{x})$ more directly. In this respect, learning explicit density models like VAEs and EBMs could be an effective solution. As we show the superior performance in both image generation and out-of-distribution detection, we believe that energy-based models, especially with discriminative representations, would be an important research direction for *reliable* generative modeling.

## REPRODUCIBILITY STATEMENT

We provide all the details to reproduce our experimental results in Appendix B. Our code is available at https://github.com/hankook/CLEL. In our experiments, we mainly use NVIDIA GTX3090 GPUs.

## ACKNOWLEDGMENTS AND DISCLOSURE OF FUNDING

This work was mainly supported by Institute of Information & communications Technology Planning & Evaluation (IITP) grant funded by the Korea government (MSIT) (No.2021-0-02068, Artificial Intelligence Innovation Hub; No.2019-0-00075, Artificial Intelligence Graduate School Program (KAIST)). This work was partly supported by KAIST-NAVER Hypercreative AI Center.

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

## A  TRAINING PROCEDURE OF CLEL

---

**Algorithm 1** Contrastive Latent-guided Energy Learning (CLEL)

---

**Require:** a latent-variable EBM $(f_\theta, g_\theta)$, a latent encoder $h_\phi$, an augmentation distribution $\mathcal{T}$, hyperparameters $\alpha, \beta, \tau > 0$, and the stop-gradient operation $\mathtt{sg}(\cdot)$.

---

1: **for** # training iterations **do**
2:   // Construct batches $\mathcal{B}$ and $\tilde{\mathcal{B}}$
3:   Sample $\{\mathbf{x}^{(i)}\}_{i=1}^n \sim p_{\mathrm{data}}(\mathbf{x})$
4:   Sample $\{\tilde{\mathbf{x}}^{(i)}\}_{i=1}^n \sim p_\theta(\mathbf{x})$ using stochastic gradient Langevin dynamics (SGLD)
5:   $\mathbf{z}^{(i)} \leftarrow \mathtt{sg}\left(h_\phi(t^{(i)}(\mathbf{x}^{(i)}))/\|h_\phi(t^{(i)}(\mathbf{x}^{(i)}))\|_2\right), t^{(i)} \sim \mathcal{T}$
6:   $\tilde{\mathbf{z}}^{(i)} \leftarrow \mathtt{sg}\left(g_\theta(f_\theta(\tilde{\mathbf{x}}^{(i)})/\|f_\theta(\tilde{\mathbf{x}}^{(i)})\|_2)\right)$

7:   // Compute the EBM loss, $\mathcal{L}_{\mathrm{EBM}}$
8:   $\mathcal{L}_{\mathrm{EBM}} \leftarrow \frac{1}{n}\sum_{i=1}^n E_\theta(\mathbf{x}^{(i)}, \mathbf{z}^{(i)}) - E_\theta(\tilde{\mathbf{x}}^{(i)}) + \alpha \cdot (E_\theta(\mathbf{x}^{(i)})^2 + E_\theta(\tilde{\mathbf{x}}^{(i)})^2)$

9:   // Compute the encoder loss, $\mathcal{L}_{\mathrm{LE}}$
10:   $\mathbf{z}_j^{(i)} \leftarrow h_\phi(t_j^{(i)}(\mathbf{x}^{(i)})), t_j^{(i)} \sim \mathcal{T}$
11:   $\mathcal{L}_{\mathrm{LE}} \leftarrow \frac{1}{2n}\sum_{i=1}^n \sum_{j=1,2} \mathcal{L}_{\mathrm{NT\text{-}Xent}}\left(\mathbf{z}_j^{(i)}, \mathbf{z}_{3-j}^{(i)}, \{\mathbf{z}_l^{(k)}\}_{k \neq i, l \in \{1,2\}} \cup \{\tilde{\mathbf{z}}^{(i)}\}_{i=1}^n; \tau\right)$

12:   Update $\theta$ and $\phi$ to minimize $\mathcal{L}_{\mathrm{EBM}} + \mathcal{L}_{\mathrm{LE}}$
13: **end for**

---

## B  TRAINING DETAILS

**Architectures.** For the spherical latent-variable energy-based model (EBM) $f_\theta$, we use the 8-block ResNet (He et al., 2016) architectures following Du & Mordatch (2019). The details of the (a) small, (b) base, and (c) large ResNets are described in Table 8. We append a 2-layer MLP with a output dimension of 128 to the ResNet, *i.e.*, $f_\theta : \mathbb{R}^{3 \times 32 \times 32} \to \mathbb{R}^{128}$. Note that we use the small model for ablation experiments in Section 4.5. To stabilize training, we apply spectral normalization (Miyato et al., 2018) to all convolutional layers. For the projection $g_\theta$, we use a 2-layer MLP with a output dimension of 128, the leaky-ReLU activation, and no bias, *i.e.*, $g_\theta(\mathbf{u}) = W_2\sigma(W_1\mathbf{u}) \in \mathbb{R}^{128}$. For the latent encoder $h_\phi$, we simply use the CIFAR variant of ResNet-18 (He et al., 2016), followed by a 2-layer MLP with a output dimension of 128.

Table 8: Our EBM $f_\theta$ architectures. For our large model, we build three independent ResNets and resize an input image $\mathbf{x} \in \mathbb{R}^{3 \times 32 \times 32}$ to three resolutions: $32 \times 32$, $16 \times 16$, and $8 \times 8$. We use each ResNet for each resolution image, concatenate their output features, and then compute the final output feature $f_\theta(\mathbf{x}) \in \mathbb{R}^{128}$ using single MLP.

| | Small | | Base | | Large | |
|---|---|---|---|---|---|---|
| Input | $(3, 32, 32)$ | | $(3, 32, 32)$ | | $(3, 32, 32), (3, 16, 16), (3, 8, 8)$ | |
| EBM $f_\theta(\mathbf{x})$ | $\mathtt{Conv}(3 \times 3, 64)$ | | $\mathtt{Conv}(3 \times 3, 128)$ | | $\mathtt{Conv}(3 \times 3, 256)$ | |
| | $\mathtt{ResBlock}(64)$ | $\times 1$ | $\mathtt{ResBlock}(128)$ | $\times 2$ | $\mathtt{ResBlock}(256)$ | $\times 2$ |
| | $\mathtt{AvgPool}(2 \times 2)$ | | $\mathtt{AvgPool}(2 \times 2)$ | | $\mathtt{AvgPool}(2 \times 2)$ | |
| | $\mathtt{ResBlock}(64)$ | $\times 1$ | $\mathtt{ResBlock}(128)$ | $\times 2$ | $\mathtt{ResBlock}(256)$ | $\times 2$ |
| | $\mathtt{AvgPool}(2 \times 2)$ | | $\mathtt{AvgPool}(2 \times 2)$ | | $\mathtt{AvgPool}(2 \times 2)$ | $\times 3$ |
| | $\mathtt{ResBlock}(128)$ | $\times 1$ | $\mathtt{ResBlock}(256)$ | $\times 2$ | $\mathtt{ResBlock}(256)$ | $\times 2$ |
| | $\mathtt{AvgPool}(2 \times 2)$ | | $\mathtt{AvgPool}(2 \times 2)$ | | $\mathtt{AvgPool}(2 \times 2)$ | |
| | $\mathtt{ResBlock}(128)$ | $\times 1$ | $\mathtt{ResBlock}(256)$ | $\times 2$ | $\mathtt{ResBlock}(256)$ | $\times 2$ |
| | $\mathtt{GlobalAvgPool}$ | | $\mathtt{GlobalAvgPool}$ | | $\mathtt{GlobalAvgPool}$ | |
| | $\mathtt{MLP}(128, 2048, 128)$ | | $\mathtt{MLP}(256, 2048, 128)$ | | $\mathtt{Concat} \to \mathtt{MLP}(768, 2048, 128)$ | |

**Training.** For the EBM parameter $\theta$, we use Adam optimizer (Kingma & Ba, 2015) with $\beta_1 = 0$, $\beta_2 = 0.999$, and a learning rate of $10^{-4}$. We use the linear learning rate warmup for the first 2k training iterations. For the encoder parameter $\phi$, we use SGD optimizer with a learning rate of $3 \times 10^{-2}$, a weight decay of $5 \times 10^{-4}$, and a momentum of 0.9 as described in Chen & He (2020).

For all experiments, we train our models up to 100k iterations with a batch size of 64, unless otherwise stated. For data augmentation $\mathcal{T}$, we follow Chen et al. (2020), *i.e.*, $\mathcal{T}$ includes random cropping, flipping, color jittering, and color dropping. For hyperparameters, we use $\alpha = 1$ following Du & Mordatch (2019), and $\beta = 0.01$ (see Section 4.5 for $\beta$-sensitivity experiments). For our large model, we use a large batch size of 256 only for learning the contrastive encoder $h_\phi$. After training, we utilize exponential moving average (EMA) models for evaluation.

**SGLD sampling.** For each training iteration, we use 60 SGLD steps with a step size of 100 for sampling $\tilde{\mathbf{x}} \sim p_\theta$. Following Du et al. (2021), we apply a random augmentation $t \sim \mathcal{T}$ for every 60 steps. We also use a replay buffer with a size of 10000 and a resampling rate of 0.1% for maintaining diverse samples (Du & Mordatch, 2019). For evaluation, we run 600 and 1200 SGLD steps from uniform noises for our base and large models, respectively.

## C    IMPLEMENTATION DETAILS FOR BYOL AND MAE

We here provide implementation details for replacing SimCLR (Chen et al., 2020) with BYOL (Grill et al., 2020) and MAE (He et al., 2021) under our CLEL framework as shown in Section 4.5.

**BYOL.** Since BYOL also learns its representations on the unit sphere, the method can be directly incorporated with our CLEL framework.

**MAE.** Since MAE's representations do not lie on the unit sphere, we incorporate MAE into our CLEL framework by the following procedure:

1. Pretrain a MAE framework and remove its MAE decoder. To this end, we simply use a publicly-available checkpoint of the ViT-tiny architecture.

2. Freeze the MAE encoder parameters and construct a learnable 2-layer MLP on the top of the encoder.

3. Train only the MLP via contrastive representation learning *without data augmentations* using our objective (13) for the latent encoder.

# D    TRAINING STABILITY WITH NORM-DIRECTION SEPARATION

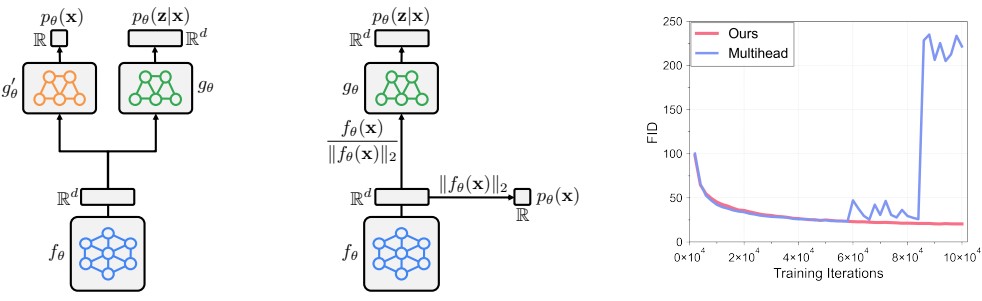

(a) Multi-head architecture     (b) Norm-direction separation     (c) FID scores during training

Figure 5: (a) A multi-head architecture design and (b) our norm-direction separation scheme for modeling $p(\mathbf{x})$ and $p(\mathbf{z}|\mathbf{x})$. (c) FID scores with various energy design choices.

At the early stage of our research, we first tested a multi-head architecture for modeling $p(\mathbf{x})$ and $p(\mathbf{z}|\mathbf{x})$. To be specific, $E(\mathbf{x}) = g'_\theta(f_\theta(\mathbf{x})) \in \mathbb{R}$ and $E(\mathbf{z}|\mathbf{x}) = \mathbf{z}^\top g_\theta(f(\mathbf{x}))$ where $f$ is a shared backbone and $g$ and $g'$ are separate 2-layer MLPs, as shown in Figure 5a. We found that, with this choice, learning $p(\mathbf{x})$ causes a mode collapse for $f_\theta(\mathbf{x})$ because all samples should be aligned with a specific direction. In contrast, learning $p(\mathbf{z}|\mathbf{x})$ encourages $f_\theta(\mathbf{x})$ to be diverse due to contrastive learning. Namely, modeling $p(\mathbf{x})$ and $p(\mathbf{z}|\mathbf{x})$ with the multi-head architecture makes some conflict during optimization. We empirically observe that the multi-head architecture is unstable in EBM training, as shown in Figure 5c. To remove such a conflict, we design the norm-direction separation for modeling p(x) and p(z|x) simultaneously (i.e., Figure 5b), which leads to training stability, as shown in Figure 5c.

# E    THE ROLE OF DIRECTIONAL PROJECTOR

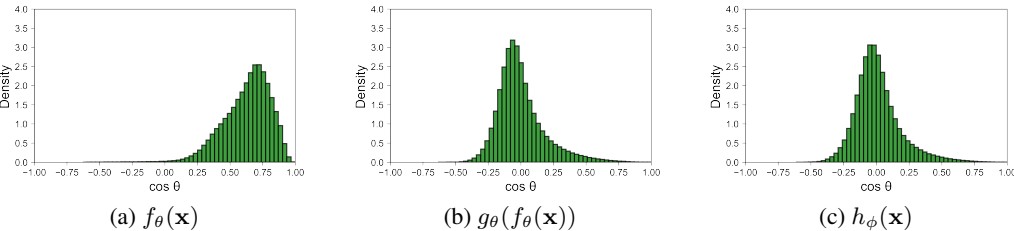

(a) $f_\theta(\mathbf{x})$       (b) $g_\theta(f_\theta(\mathbf{x}))$       (c) $h_\phi(\mathbf{x})$

Figure 6: Cosine similarity distributions using (a) $f_\theta(\mathbf{x})$, (b) $g_\theta(f_\theta(\mathbf{x}))$, and (c) $h_\phi(\mathbf{x})$ features.

Our directional projector $g_\theta$ is designed for narrowing the gap between the EBM feature direction $f_\theta(\mathbf{x})/\|f_\theta(\mathbf{x})\|_2$ and the "uniformly-distributed" latent variable $p_{\text{data}}(\mathbf{z})$ (i.e., $h_\theta(\mathbf{x})/\|h_\theta(\mathbf{x})\|_2$). Specifically, the contrastive latent variable $p_{\text{data}}(\mathbf{z})$ is known to be uniformly distributed (Wang & Isola, 2020), but we observed that it is difficult to optimize the feature direction $f_\theta(\mathbf{x})/\|f_\theta(\mathbf{x})\|_2$ to be uniform along with learning our norm-based EBM $p_\theta(\mathbf{x}) \propto \exp(-\|f_\theta(\mathbf{x})\|_2^2)$ at the same time. As empirical supports, we analyze the cosine similarity distributions using $f_\theta(\mathbf{x})$, $g_\theta(f_\theta(\mathbf{x}))$, and $h_\phi(\mathbf{x})$ features on CIFAR-10, as shown in Figure 6. This figure shows that $f_\theta(\mathbf{x})$ tends to learn similar directions (see Figure 6a) while $h_\phi(\mathbf{x})$ tends to be uniformly distributed (see Figure 6c). Hence, it is necessary to employ a projection between them. We found that our projector $g_\theta$ successfully narrows the gap as shown in Figure 6b, which significantly improves training EBMs.

# F  FLEXIBILITY OF NORM-BASED ENERGY FUNCTION

Our norm-based energy parametrization does not sacrifice the flexibility compared to the vanilla parametrization of EBMs. We here show that any vanilla EBM $p_1(\mathbf{x}) \propto \exp(f_1(\mathbf{x}))$, $f_1(\mathbf{x}) \in \mathbb{R}$, can be formulated by a norm-based EBM $p_2(\mathbf{x}) \propto \exp(\|f_2(\mathbf{x})\|_2^2)$, $f_2(\mathbf{x}) \in \mathbb{R}^d$, on a compact input space $\mathcal{X}$ (e.g., an image $\mathbf{x}$ lies on the continuous pixel space $\mathcal{X} = [0, 255]^{HWC}$).

Let $b := \min_{\mathbf{x} \in \mathcal{X}} f_1(\mathbf{x})$ be the minimum value of $f_1$. Then, $p_3(\mathbf{x}) \propto \exp(f_1(\mathbf{x}) - b)$ is identical to $p_1$ due to the normalizing constant. Furthermore, $p_3$ can be formulated as a special case of the norm-based EBM $p_2$: for example, if the first component of $f_2$ is the same as $\sqrt{f_1(\mathbf{x}) - b}$ and other components are zero, then $p_1$ and $p_2$ model the exactly same distribution. Therefore, energy-modeling with our norm-based design is not much different from that with the vanilla form.

