# OpenReview forum: "Guiding Energy-based Models via Contrastive Latent Variables"
_ICLR.cc/2023/Conference — ICLR 2023 notable top 25%_

### Official Review · Reviewer_hpmQ · 2022-10-25

**Confidence:** 4
**Correctness:** 3
**Technical Novelty And Significance:** 3
**Empirical Novelty And Significance:** 3
**Recommendation:** 6

**Clarity, Quality, Novelty And Reproducibility:**

The paper is generally clear and I think the structure starting with preliminaries makes the reading much easier. The authors provided the code and it looks good to me. Again, though the paper sticks to the raw EBM framework, the contrastive module is embedded neatly. I think this could potentially benefit the area.

**Strength And Weaknesses:**

Pros:
1. This is a very interesting idea to embed contrastive learning into EBM. Indeed, contrastive learning has been widely used in a lot of scenarios. Employing it in EBM is quite novel.
2. In experiments, the paper demonstrates unconditional generation, conditional generation, out-of-sampling detection and conditional sampling. It's thus convincing that this work is versatile, and potentially can be used in many scenarios.

Cons:
1. Could you further explain the P in Eq. 9?
2. The directional projector lacks explanation or intuitions. It's better to explain the motivation and some insights behind this module.
3. I think one thing you claim in your paper is that your method can make the EBM training more stable. Maybe you can give more examples to illustrate that.
4. Some numbers are still worse than GAN-based numbers. Do you have any insights or ideas?

**Summary Of The Paper:**

This paper proposes a very neat idea of embedding contrastive learning into energy-based model learning. On top of the normal latent-variable energy-based models, the authors introduce augmentation and contrastive representation learning to regularize the latent space. The general idea is to still keep the empirical KL divergence, regularized by energy regularization terms. Besides, this work adds contrastive learning, inspired by SimCLR, on the latent space modeling. The experiments give both qualitative and quantitative results. This work can generally outperform other energy or autoencoder-based methods.

**Summary Of The Review:**

This is a good idea with many experimental validations. Some claims need to be further supported. And the gap with other type of models should be better explained. Some modules should be expanded to further facilitate understanding.

---

> ### Author Response · Authors · 2022-11-14
> **Response to Reviewer hpmQ**
>
> Dear Reviewer hpmQ,
>
> We sincerely thank you for your helpful feedback and insightful comments. We address your comments and questions below. In the revised draft, we mark our revisions in “$\color{blue}\text{blue}$”.
>
> ---
>
> **[Q1]** Could you further explain the P in Eq. 9? \
> **[A1]** Thank you for the constructive question for clarity. Eq (9) is equivalent to a two-step sampling procedure as follows: (i) sample an augmentation $t\sim T$, and then (ii) compute the representation $z$ using the contrastive encoder $h_\phi$. We modified the notation and Eq. (9) for clarity in the revision.
>
> ---
>
> **[Q2]** The directional projector lacks explanation or intuition. \
> **[A2]** Thank you for the constructive feedback. Our directional projector $g$ is designed for narrowing the gap between the EBM feature direction $f(x)/\lVert f(x)\rVert$ and the “uniformly-distributed” latent variable $p_\text{data}(z)$. Specifically, the contrastive latent variable $p_\text{data}(z)$ is known to be uniformly distributed [1], but we observed that it is difficult to optimize the feature direction $f(x)/\lVert f(x)\rVert$ to be uniform along with learning our norm-based EBM $p(x)\propto\exp(-\lVert f(x)\rVert^2)$ at the same time. So we added the directional projector, and we found that this significantly improves training EBMs. We added this explanation with empirical supports into our revision (Appendix E).
>
> [1] Wang and Isola, Understanding Contrastive Representation Learning through Alignment and Uniformity on the Hypersphere, ICML 2020
>
> ---
>
> **[Q3]** Can give more examples to illustrate how the method improves the EBM training stability. \
> **[A3]** Thank you for the constructive feedback. At the early stage of our research, we first tested a multi-head architecture for modeling $p(x)$ and $p(z|x)$. To be specific, $E(x)=g'_\theta(f_\theta(x))\in\mathbb{R}$ and $E(z|x)=z^\top g_\theta(f(x))$ where $f$ is a shared backbone and $g$ and $g'$ are separate MLPs. We found that, with this choice, learning $p(x)$ causes a mode collapse for $f(x)$ because all samples should be aligned with a specific direction. In contrast, learning $p(z|x)$ encourages $f(x)$ to be diverse due to contrastive learning. Namely, modeling $p(x)$ and $p(z|x)$ with the multihead architecture makes some conflict (i.e., training instability) during optimization. To remove such a conflict, we design the norm-direction separation for modeling $p(x)$ and $p(z|x)$ simultaneously, which makes training stable. We added this discussion with supporting experiments into the revision (Appendix D).
>
> ---
>
> **[Q4]** Some numbers are still worse than GAN-based numbers. Do you have any insights or ideas? \
> **[A4]** We think EBMs are often worse than GANs in terms of generation since (i) the former aims to learn the explicit density while the latter does not, and (ii) the latter utilizes a generative network while the former does not. In recent years, there have been several attempts to incorporate auxiliary generators into EBMs like GANs, e.g., [1]. We strongly believe that our framework can be also further improved by incorporating an auxiliary generator, and it would be an interesting research direction.
>
> [1] Arbel et al., Generalized Energy Based Models, ICLR 2021

---

### Official Review · Reviewer_zYAi · 2022-10-25

**Confidence:** 5
**Clarity, Quality, Novelty And Reproducibility:** The paper is well-written, and the id…
**Correctness:** 4
**Technical Novelty And Significance:** 3
**Empirical Novelty And Significance:** 3
**Recommendation:** 8

**Strength And Weaknesses:**

The idea is novel and experimentally they show a great performance (w.r.t. to FID score) on CIFAR-10.  Although the performance is not better than the SOT in the field, it is impressive regarding other EBMs training methods. There are many applications in that EBMs have an advantage over other probabilistic methods such as score models, so better EBM training algorithms are important to the community.

The main weakness of the paper is that the authors didn't justify the joint energy model that they are using.
For example, they say that "the remaining information in $\|| f_\theta(x)\||_2$ is used for modeling $p_\theta(x)$. What do you mean by that?
or when you say that "we separate f into direction and norm and we found that this separation reduces the conflict between $p_\theta$ and p\theta(z|x) optimization" what exactly do you mean? Are the direction of the gradient inconsistent? What else did you try?
Why does only a particular form of g work well? Why does using the identity function as g drastically reduce the performance of the model? The FID score when using the identity function is twice higher as using beta=0!

In Table 5, setting b does not use the negative example from p_theta for training the encoder. Does this mean that one can use a pre-trained encoder and distill the representation knowledge into E(x,z)? Have you tried something like that?


Typo: "into our EMB training"

**Summary Of The Paper:**

The paper proposes a latent variable energy-based model by introducing an energy form for the joint model p(x,z).
Intuitively, they augment data point x with z that comes from a separate encoder (that is trained using conservative representation learning) and train the joint energy model using contrastive divergence. They also use the z that minimizes E(x,z) as a negative sample to train the encoder.
This setup enforces that the energy model assigns lower energy to the latent representation that is aligned with the external decoder.

The authors show the capability of their methods using experiments on OOD detection, conditional and unconditional generation, and compositional generation.

**Summary Of The Review:**

The paper is relevant to the community, novel, and shows an interesting connection between SSL representation learning and EBM training.
The experimental results are strong and the authors did an extensive analysis of their methods. The main weakness is the lack of sufficient discussion regarding the used energy form and ablation study. For example, the authors show that the identity function is not a useful projector but not discussing why?

---

> ### Author Response · Authors · 2022-11-14
> **Response to Reviewer zYAi**
>
> Dear Reviewer zYAi,
>
> We sincerely thank you for your helpful feedback and insightful comments. We address your comments and questions below. In the revised draft, we mark our revisions in “$\color{blue}\text{blue}$”.
>
> ---
>
> **[Q1]** Justification for our norm-direction separation for modeling p(x) and p(z|x). \
> **[A1]** Thank you for the constructive feedback. At the early stage of our research, we first tested a multi-head architecture for modeling $p(x)$ and $p(z|x)$. To be specific, $E(x)=g'_\theta(f_\theta(x))\in\mathbb{R}$ and $E(z|x)=z^\top g_\theta(f(x))$ where $f$ is a shared backbone and $g$ and $g'$ are separate MLPs. We found that, with this choice, learning $p(x)$ causes a mode collapse for $f(x)$ because all samples should be aligned with a specific direction. In contrast, learning $p(z|x)$ encourages $f(x)$ to be diverse due to contrastive learning. Namely, modeling $p(x)$ and $p(z|x)$ with the multihead architecture makes some conflict (i.e., training instability) during optimization. To remove such a conflict, we design the norm-direction separation for modeling $p(x)$ and $p(z|x)$ simultaneously, which makes training stable. We added this discussion with supporting experiments into the revision (Appendix D).
>
> ---
>
> **[Q2]** Why does only a particular form of g work well? Why does using the identity function as g drastically reduce the performance of the model? \
> **[A2]** Thank you for the important question. Our directional projector $g$ is designed for narrowing the gap between the EBM feature direction $f(x)/\lVert f(x)\rVert$ and the “uniformly-distributed” latent variable $p_\text{data}(z)$. Specifically, the contrastive latent variable $p_\text{data}(z)$ is known to be uniformly distributed [1], but we observed that it is difficult to optimize the feature direction $f(x)/\lVert f(x)\rVert$ to be uniform along with learning our norm-based EBM $p(x)\propto\exp(-\lVert f(x)\rVert^2)$ at the same time. So we used such a MLP-based projector rather than the identify function, and we found that this significantly improves training EBMs. We added this explanation with empirical supports into our revision (Appendix E).
>
> [1] Wang and Isola, Understanding Contrastive Representation Learning through Alignment and Uniformity on the Hypersphere, ICML 2020
>
> ---
>
> **[Q3]** One can use a pre-trained encoder and distill the representation knowledge into E(x,z)? Have you tried something like that? \
> **[A3]** Yes, we have tried. When using the representation learned by MAE [1], we used the pretrained MAE encoder and distilled its representation into our E(x,z) (see Appendix C for details). We found that this also improves the generation quality while joint-training with contrastive learning was most effective in our experiments as shown in Table 7. However, we think distilling knowledge from any pre-trained encoder into EBMs would be an interesting research direction.
>
> [1] He et al., Masked Autoencoders Are Scalable Vision Learners, 2021
>
> ---
>
> **[Q4]** Typo \
> **[A4]** Thank you for pointing this out! We corrected the typo in the revision.

---

### Official Review · Reviewer_nYLf · 2022-11-02

**Confidence:** 4
**Correctness:** 3
**Technical Novelty And Significance:** 2
**Empirical Novelty And Significance:** 3
**Recommendation:** 5

**Clarity, Quality, Novelty And Reproducibility:**

This paper is generally clear. The reproducibility can be guaranteed if the codes provided work properly.
About the novelty and quality, incorporating latent variable is important in statistical models. The authors have demonstrated a heuristic way to obtain the latent variable by contrastive learning, which shows the improvement in vision tasks (generation quality and OOD detection). Due to the heuristic motivation, it is better to provide more insights about the introduction of the latent variable beyond the downstream performance, which are better justifications for CLEL algorithms.

**Strength And Weaknesses:**

Strength:

The introduction of latent variable is important in statistical inference tasks. It is exciting to see that the latent variables help EBM in practice.

I appreciate that the endeavor of the authors to adapt EBM models to visual datasets, and the resulting generation quality and OOD detection have been improved over previous EBMs.


Weakness:

Although the introduction of latent variable is important, the CLEL just combine contrastive learning and EBM in a brute way, which is intuitive but incremental. What I am concerned about the most is the significance and novelty of this work, which is fully heuristic and direct in my opinion.  It is good to see that CLEL can improve image generation and OOD detection but I will really appreciate it if you can provide more insights about the “latent variables” rather than the downstream performance (which is also not strong enough). I must say that the paper can be significantly improved if it can provide enough evidence to show that what is the “latent variable learned by contrastive learning” and provide more statistical logics about the benefits.


The authors claim that the proposed model benefits from the introduction of true underlying latent variable. I believe that it is necessary to justify the why it is the true latent variable? Does it have explainable representations? Or can you prove that in what sense it is true.

I must say the generation quality is not the core issue of EBM’s training, thus I don’t think improving generation quality is significant enough, since we have many other choices.

**Summary Of The Paper:**

This paper proposes to improve EBM training via contrastive representation learning (CRL).

In particular, it uses SimCLR framework to train the latent variable and implement latent-variable EBM with joint probability. The empirical results show that the proposed framework is more efficient and achieves lower FID scores and better OOD detection on visual datasets.

**Summary Of The Review:**

In general, I think CLEL is a decent but incremental empirical work that verifies the benefits of contrastive latent variables in EBM training. The claims are justified by the downstream tasks and some ablation studies. But I do believe that there should be some results beyond the heuristic motivation and plain numerical results to justify this work so that it can be significant enough to convince the community. Thus, I encourage the authors to repackage the work and find out more merits about CLEL.

---

> ### Author Response · Authors · 2022-11-14
> **Response to Reviewer nYLf**
>
> Dear Reviewer nYLf,
>
> We sincerely thank you for your helpful feedback and insightful comments. We address your comments and questions below. In the revised draft, we mark our revisions in “$\color{blue}\text{blue}$”.
>
> ---
>
> **[Q1]** Could you provide more insights about the introduction of the contrastive latent variable? \
> **[A1]** Thank you for the constructive feedback on our framework. As we mentioned in the draft, our intuition behind the contrastive latent variables is two-fold: (i) latent variables containing some "good" contexts of data can improve generative models, and (ii) the contrastive representation encodes such good contexts.
>
> In practice, conditional generative modeling given some good contexts of data, e.g., p(x|y) or p(x|z), is known to be much easier than unconditional modeling p(x). For example, labels [1-2] or rich information such as captions [3] can improve the generation quality. In addition, the mode collapse problem can be resolved by learning the label p(y|x) [4] or the manifold space p(z|x) [5]. Since our framework implicitly learns both p(x|z) and p(z|x) via modeling the joint distribution p(x,z), our EBM training can benefit from the latent variables.
>
> The remaining question is why the representation learned by contrastive learning can be considered good latent variables. First, the contrastive representation has already improved the performances of various downstream tasks [6]. Furthermore, contrastive learning is theoretically based on InfoNCE [7], which maximizes mutual information I(X;C) between the data X and the context C (the latent variable Z in our formulation). Namely, Z can contain enough information about X with theoretical support. Based on these observations, we believe contrastive representation contains good enough context of data so that it can improve EBM performances. Table 7 indeed confirms that the contrastive representation is the best latent variable among other self-supervised representations (e.g., BYOL, MAE).
>
> We incorporated this discussion into our revision (Section 3).
>
> [1] Mirza and Osindero, Conditional Generative Adversarial Nets, 2014 \
> [2] Oord et al., Conditional Image Generation with PixelCNN Decoders, 2016 \
> [3] Reed et al., Generative adversarial text-to-image synthesis, 2016 \
> [4] Odena et al., Conditional Image Synthesis With Auxiliary Classifier GANs, ICML 2017 \
> [5] Bang and Shim, MGGAN: Solving Mode Collapse Using Manifold-Guided Training, ICCV Workshops, 2021 \
> [6] Chen et al., A Simple Framework for Contrastive Learning of Visual Representations, ICML 2020 \
> [7] Oord et al., Representation Learning with Contrastive Predictive Coding, 2018
>
> ---
>
> **[Q2]** The generation quality is not the core issue of EBM’s training, thus I don’t think improving generation quality is significant enough, since we have many other choices. \
> **[A2]** We politely disagree with the reviewer’s opinion as Reviewer zYAi highlighted: *“there are many applications in that EBMs have an advantage over other probabilistic methods such as score models, so better EBM training algorithms are important to the community”*. For example, unlike other generative models, EBMs can generate samples under compositionality of multiple concepts (see Section 4.4), which has been utilized in not only computer vision [1], but also chemistry [2]. Out-of-detection (Section 4.2) and conditional generation (Section 4.3) are other applications of our EBM models. Furthermore, since EBM is closely related to GANs [3] and score-based models [4-5], we think our idea can be utilized for improving other generative models in the future. Hence, we strongly believe that our work makes a meaningful contribution to the EBM and generative modeling communities.
>
> [1] Du et al., Compositional Visual Generation with Energy Based Models, NeurIPS 2020, \
> [2] Liu et al., GraphEBM: Molecular Graph Generation with Energy-Based Models, ICLR 2021 Workshop on EBMs \
> [3] Che et al., Your GAN is Secretly an Energy-based Model and You Should use Discriminator Driven Latent Sampling, NeurIPS 2020 \
> [4] Song and Kingma, How to Train Your Energy-Based Models, 2021 \
> [5] Gao et al., Learning Energy-Based Models by Diffusion Recovery Likelihood, ICLR 2021

---

> > ### Comment · Reviewer_nYLf · 2022-11-18
> > **Response**
> >
> > Thank you for your response. I really appreciate your references to compare with other conditional generation works. Unfortunately, it is still opaque for me about how “conditions” you use help the generation from a statistical view (though it is reasonable from a heuristic	view). The most paper you mentioned, the “condition” means the data are given with labels or other information (such as texts). The task is totally different, because they aim at performing controllable generation but this paper claims to “improve” general generation with the unsupervised representation. Moreover, the unsupervised information you use, (representation), should also be directly obtained in a generative model. For example, the representation in GANs are also good for downstream tasks [1]. However, you use another unsupervised representation (contrastive learning) to improve the EBM training, it is not clear from my perspective. Why does the intrinsic representation in EBM is not good enough? How did the contrastive representation helps EBM training? These ambiguities are really important for me to evaluate the significance of this work.
> >
> > Besides, about the generation quality, I appreciate the efforts of the authors to improve the EBM’s image generation. I understand that EBMs have some merits against other generative models. However, you should make justifications for the merits rather than just performing experiments on the vision tasks and giving FID results (OOD detection is a good supplement but not strong enough so far). For example, in [2], it emphasize the conceptual learning of EBMs and make justifications for conceptual advantages. Indeed, EBM is closely related to GANs and score-based models, but I do not think current justifications provide enough information for readers to understand the principle behind it, not just heuristic motivations. Moreover, they perform better in image generation tasks. If your idea really works on their frameworks, why not just improve them to obtain a SOTA generation result. Why do you choose EBM with the contrastive representations? Why do you choose to improve a relative weak model (in visual generation)? If it is special, you have to clarify the merits; if it is not special, you can choose the best visual generative model to get a SOTA, which is uncontroversial and impressive.
> >
> > Overall:
> > I admit that there are great potential of this work and I really appreciate the authors efforts to make EBM better on visual tasks. Given the heuristic methodology (without any theoretical insight) and the indirect empirical justification, it is not ready for publication at this moment from my perspective. Thus, I cannot give a higher rating at the moment.
> >
> >
> >
> > [1] Donahue, J., & Simonyan, K. (2019). Large scale adversarial representation learning. Advances in neural information processing systems, 32.
> >
> > [2] Du et al., Compositional Visual Generation with Energy Based Models, NeurIPS 2020.

---

> > > ### Author Response · Authors · 2022-11-24
> > > **Additional Response to Reviewer nYLf (2/2)**
> > >
> > > **[Q2] Why we choose EBMs** (response to the second paragraph)
> > >
> > > We first remark that our paper already demonstrated the merits of our model in Section 4.3 and Section 4.4. Specifically, we provide instance-conditional sampling (Section 4.3) and compositional sampling (Section 4.4) using only single EBM without explicit conditional learning. This is not available in existing EBMs, e.g., [1] requires multiple independently-trained conditional EBMs for compositional sampling.
> > >
> > > The reason why we choose EBMs with the contrastive representations rather than other generative models is that EBMs are more favorable to design the joint energy $E_\theta(\mathbf{x},\mathbf{z})$. In contrast, it is non-trivial to design a joint density $p_\theta(\mathbf{x},\mathbf{z})$ for other generative models such as score-based models. Moreover, training SotA models often requires huge computational resources (e.g., DDPM [2] requires 8 V100 GPUs), so we are not able to conduct such experiments due to our limited resources. Hence, in this work, we focused on EBMs and we left the application of our idea to other generative models for future work.
> > >
> > > We also emphasize that improving EBMs is an important research problem itself as it has been actively studied, e.g., [3-7]. We politely ask you to consider our contribution under perspectives of the EBM community and our potential for generative modeling.
> > >
> > > [1] Du et al., Compositional Visual Generation with Energy Based Models, NeurIPS 2020 \
> > > [2] Ho et al., Denoising Diffusion Probabilistic Models, NeurIPS 2020 \
> > > [3] Xie et al., A Tale of Two Flows: Cooperative Learning of Langevin Flow and Normalizing Flow Toward Energy-Based Model, ICLR 2022 \
> > > [4] Wang et al., A Unified Contrastive Energy-based Model for Understanding the Generative Ability of Adversarial Training, ICLR 2022 \
> > > [5] Nijkamp et al., MCMC Should Mix: Learning Energy-Based Model with Neural Transport Latent Space MCMC, ICLR 2022 \
> > > [6] Kan et al., Bi-level Doubly Variational Learning for Energy-based Latent Variable Models, CVPR 2022 \
> > > [7] Yin et al., Learning Energy-Based Models With Adversarial Training, ECCV 2022

---

> > > ### Author Response · Authors · 2022-11-24
> > > **Additional Response to Reviewer nYLf (1/2)**
> > >
> > > Thank you for further feedback on our paper. In what follows, we address your concerns one by one.
> > >
> > > ---
> > >
> > > **[Q1] More underlying insights for our method** (response to the first paragraph)
> > >
> > > As you mentioned, representation learning is closely related to generative modeling, e.g., both GANs [1] and EBMs [2] have intrinsic representations. However, the intrinsic representations of EBM are worse than contrastive representations and they can be improved by guidance from such a better representation (like knowledge distillation). Our main idea is indeed improving intrinsic representations in EBMs via such a guidance, which would result in better EBM models. For example, the representation of the EBM baseline (i.e., without guidance) achieves only 56% accuracy under linear evaluation in CIFAR-10 while our EBM representation does 77% thanks to the guidance via contrastive learning. This empirically supports that our approach significantly improves the EBM’s intrinsic representations during EBM training, which is the first intuition why our method works.
> > >
> > > Second, our framework aims to learn the joint distribution $p_\text{data}(x,z)$ which can be decomposed by the conditional distribution $p_\text{data}(x|z)$ and the marginal latent distribution $p_\text{data}(z)$. Namely, our framework is closely related to conditional models and can benefit from their advantages as we described in the previous response. Similarly, Joint Energy-based Models (JEMs) [3] learn the joint distribution $p_\theta(x,y)$ and successfully generate better unconditional samples via $p_\theta(x)$ like ours (see Appendix B of [3]). LSGM [4] also benefits from modeling $p_\theta(x|z)$ and $p_\theta(z)$ separately for unconditional generation. We think such a connection between conditional and unconditional modeling provides another angle why our method works.
> > >
> > > Finally, we also provide a theoretical insight to address your concern. Since EBMs based on deep neural networks are hard to be theoretically analyzed, we here consider a simple EBM to provide some intuition of our method: a restricted Boltzmann machine (RBM) with data variables $\mathbf{x}\in\\{0,1\\}^n$ and latent variables $\mathbf{z}\in\\{0,1\\}^m$. For learning $p_\text{data}(\mathbf{x})$, the best-known learning algorithm has $O(n^d)$ time-complexity where $d$ is the maximum degree of the latent variables [5-7]. However, if latent variables are additionally observed (i.e., learning $p_\text{data}(\mathbf{x},\mathbf{z})$), the time-complexity reduces to $O((n+m)^2)$ [8]. This is because observing latent variables provides conditional independence structure between data variables, i.e., it simplifies the dependency structure of the data variables. Similarly, contrastive latent variables contain semantic information of input images, so we think our approach could simplify the data structure by observing the latent variables. We believe a more comprehensive theoretical study would be a meaningful future research direction.
> > >
> > > [1] Donahue & Simonyan, Large Scale Adversarial Representation Learning, NeurIPS 2019 \
> > > [2] Wu et al., Conjugate Energy-Based Models, ICML 2021 \
> > > [3] Grathwohl et al., Your Classifier is Secretly an Energy Based Model and You Should Treat it Like One, ICLR 2020 \
> > > [4] Vahdat et al., Score-based Generative Modeling in Latent Space, NeurIPS 2021 \
> > > [5] Hamilton et al., Information Theoretic Properties of Markov Random Fields, and their Algorithmic Applications, NeurIPS, 2017 \
> > > [6] Klivans & Meka, Learning graphical models using multiplicative weights, IEEE Annual Symposium on Foundations of Computer Science (FOCS), 2017 \
> > > [7] Vuffray et al., Efficient learning of discrete graphical models, 2019 \
> > > [8] Bresler, Efficiently learning Ising models on arbitrary graphs, 2014

---

### Official Review · Reviewer_eaGu · 2022-11-03

**Confidence:** 5
**Correctness:** 3
**Technical Novelty And Significance:** 4
**Empirical Novelty And Significance:** 3
**Recommendation:** 8

**Clarity, Quality, Novelty And Reproducibility:**

Clarity: the paper is clearly written and easy to follow.

Quality: the formulation is technically sound to the best of my knowledge. Experimental results are convincing.

Novelty: the proposed jointly training method of EBM and CRL encoder is novel. The parametrization of latent-variable EBM is also novel.

Reproducibility: implementation details are adequately provided.

**Strength And Weaknesses:**

### Strength
- The paper presents an interesting way to parametrize latent-variable EBMs, such that $p(x)$ and $p(z|x)$ are in closed-form.
- Given the proposed model formulation, the paper proposes a wise way to combine it with contrastive representation learning. Results indicate that such combination can boost the performance of EBMs reasonably well.

### Weaknesses and questions
- The parametrization of $E_\theta(x) = \frac{1}{2} \|f_\theta(x)\|_2^2 + C$ seems to be a constrained set of functions. I'd like to see more discussions in terms of why it could serve as a general set for modeling energy functions.
- I don't find information in the paper about whether the SimCLR encoder is pretrained or jointly estimated with the EBM starting from scratch. If it is pretrained, I'd like to see another ablation where the encoder is fixed as the pretrained one and only EBM is learned.
- In the ablation study, all model have only been updated for 50k iterations. Do all the models really converge with such a small amount of iterations?
- Given the proposed framework also provides a new objective for the encoder of contrastive representation learning, does it also lead to better learned features $z$ compared to the original CRL in terms of downstream tasks such as classification?

**Summary Of The Paper:**

This paper proposes to improve the performance of energy-based models by incorporating latent variables to the model, assuming the positive samples of which come from an encoder optimized by contrastive representation learning. The latent variable EBM is parametrized in a way such that $p(x)$ and $p(z|x)$ can be easily separated. The performance can be further improved by updating the encoder with sampled $z$ from the EBM as auxiliary negative samples in CRL. Experimental results show it leads to state-of-the-art generative performance within EBMs. The learned model is also capable of conditional and compositional generation.

**Summary Of The Review:**

The paper proposes an interesting way for parametrizing latent-variable EBM and combining it with CRL. Experimental results are convincing that it is a promising approach to improving EBMs. It will be even strong if the paper could show the joint estimation method can also improve the learned features of the encoder.

---

> ### Author Response · Authors · 2022-11-14
> **Response to Reviewer eaGu**
>
> Dear Reviewer eaGu,
>
> We sincerely thank you for your helpful feedback and insightful comments. We address your comments and questions below. In the revised draft, we mark our revisions in “$\color{blue}\text{blue}$”.
>
> ---
>
> **[Q1]** The norm-based energy parametrization seems to be a constrained set of functions. \
> **[A1]** This is not true, i.e., our norm-based energy parametrization does not sacrifice the flexibility compared to the vanilla parametrization of EBMs. To be specific, one can show that any vanilla EBM $p_1(\mathbf{x})\propto\exp(f_1(\mathbf{x}))$, $f_1(\mathbf{x})\in\mathbb{R}$, can be formulated by a norm-based EBM $p_2(\mathbf{x})\propto\exp(\lVert f_2(\mathbf{x}) \rVert_2^2)$, $f_2(\mathbf{x})\in\mathbb{R}^d$, on a compact input space $\mathcal{X}$ (e.g., an image $\mathbf{x}$ lies on the continuous pixel space $\mathcal{X}=[0,255]^{HWC}$).
>
> Let $b:=\min_{\mathbf{x}\in\mathcal{X}} f_1(\mathbf{x})$ be the minimum value of $f_1$. Then, $p_3(\mathbf{x})\propto\exp(f_1(\mathbf{x})-b)$ is identical to $p_1$ due to the normalizing constant. Furthermore, $p_3$ can be formulated as a special case of the norm-based EBM $p_2$: for example, if the first component of $f_2$ is the same as $\sqrt{f_1(\mathbf{x})-b}$ and other components are zero, then $p_1$ and $p_2$ model the exactly same distribution. Therefore, energy-modeling with our norm-based design is not inferior to that with the vanilla form with respect to the expressive power.
>
> We incorporated this discussion into the revision (Appendix F).
>
> ---
>
> **[Q2]** SimCLR encoder is pretrained or jointly estimated with the EBM starting from scratch? \
> **[A2]** Thank you for the constructive feedback to improve the clarity of our manuscript. Our SimCLR encoder is jointly optimized with the EBM starting from scratch. We emphasize this more in the revision (Section 3.3).
>
> ---
>
> **[Q3]** Do all the ablation models really converge with such a small amount of iterations (i.e., 50k iterations)? \
> **[A3]** Although our small models are not perfectly converged with 50k training iterations, the observed gains are enough to demonstrate the effectiveness of our method as shown in our ablation experiments. To further alleviate your concern, we additionally conduct an ablation experiment with 100k training iterations: we found that the FID gains of CLEL (compared to baselines) at 50k and 100k iterations are consistent, i.e., 6.73 (42.46 → 35.73) and 5.41 (36.52 → 31.11), respectively.
>
> ---
>
> **[Q4]** Does the objective also lead to better learned features z compared to the original CRL in terms of downstream tasks such as classification? \
> **[A4]** Unfortunately, we found that our new objective for the encoder of contrastive learning does not improve the downstream classification performance as noted in footnote 1. We think our approach is instead related to discrimination between in-distribution and out-of-distribution (OOD) samples as shown in our OOD detection experiments. However, it would be an interesting research direction to improve the classification performance along with the better generation quality of EBMs.

---

### Author Response · Authors · 2022-11-14
**General Response**

Dear reviewers and AC,

We sincerely appreciate your valuable time and effort spent reviewing our manuscript.

As reviewers highlighted, our work aims at an important problem (Reviewer zYAi, nYLf) with an interesting/novel method (Reviewer eaGu, zYAi, hpmQ), strong empirical results (ALL Reviewers), and a well-written/easy-to-follow writeup (ALL Reviewers).

We appreciate your constructive comments on our manuscript. In response to the comments, we have carefully revised and enhanced the manuscript with the following additional discussions and experiments:

- Additional experiment of CLEL-Large on ImageNet-32x32 (Table 2)
- Clarification about our intuition (Section 3)
- Clarification about how to sample latent variables (Eq. (9))
- Discussion about the training stability with our norm-direction separation design (Section 3.2, Appendix D)
- Discussion about our directional projector $g$ (Section 3.2, Appendix E)
- Discussion about the flexibility of our norm-based energy design (Section 3.2, Appendix F)

These updates are temporarily highlighted in “$\color{blue}\text{blue}$” for your convenience to check.

We hope our response and revision sincerely address all the reviewers’ concerns.

Thank you very much.

Best regards, \
Authors.

---

### Public Comment · ~Yihong_Luo1 · 2023-03-28
**Great Work!**

Hi, this work is impressive. I'm curious about how you guys choose the hyper-parameters. The step size 100 seems too big, is there any insight into selecting such a big step size for MCMC?

---

> ### Author Response · Authors · 2023-03-30
> **I appreciate your interest in our work.**
>
> Hi,
>
> We choose our MCMC hyperparameters following Du et al. (2021), which also utilizes the data augmentation technique for improving MCMC sampling as ours. The augmentation technique makes training stable and generation diverse, but it also causes distribution shifts, so one should use either a large number of steps or a large step size to remedy the issue.
>
> Best, \
> Hankook Lee
>
> [Du et al., 2021] Improved contrastive divergence training of energy based models, ICML 2021

---

### Decision · Program_Chairs · 2023-01-20

**Decision:**

Accept: notable-top-25%

**Justification For Why Not Higher Score:**

There are still some clarifications to be added about the theoretical motivations for using CL to train EBMs as pointed out by Reviewer nYLf.

**Justification For Why Not Lower Score:**

Using CL in the context of generative modeling is novel and can have significant implications given the interest of the high community in both SSL and generative modeling.


**Metareview: Summary, Strengths And Weaknesses:**

The paper proposes a method to train latent EBMs using contrastive learning. The empirical results show that the proposed framework is more efficient and achieves lower FID scores and better OOD detection on visual datasets.
The reviewers agree that the method is novel and could has significant implications for generative modeling.

Reviewer nYLf points out a lack of theoretical insight about the approach. I agree that additional theoretical motivation for why the approach would work would be valuable. It would be also interesting to investigate what contrastive learning brings to the latent representation compared to the latent representation learned by a standard latent EBM.


However, I think that it is still interesting to see that CL can be used to train EBMs. CL incorporates additional information (through augmentation) which is not present in standard EBM training. Showing this additional signal to be beneficial empirically is still valuable.

For these reasons, and from the other reviewers' comments, I'm inclined to accept the paper for this empirical insight and to recommend adding more discussions about the limitations/ the open questions that are not addressed by the present work as suggested by Reviewer nYLf.







**Note From Pc:**

if the above contains the word "oral" or "spotlight" please see: "oral" presentation means -> notable-top-5% and "spotlight" means -> notable-top-25%. As stated in our emails, we are disassociating presentation type from AC recommendations